# Conversion of Residual Palm Oil into Green Diesel and Biokerosene Fuels under Sub- and Supercritical Conditions Employing Raney Nickel as Catalyst

**Eduardo Falabella Sousa-Aguiar [1], Carolina Zanon Costa [1], Maria Antonieta Peixoto Gimenes Couto [1], Débora de Almeida Azevedo [2] and José Faustino Souza de Carvalho Filho [1,\*]**

[1] Escola de Química, Universidade Federal do Rio de Janeiro, Rio de Janeiro 21941-909, RJ, Brazil; efalabella@eq.ufrj.br (E.F.S.-A.); zanoncarol@gmail.com (C.Z.C.); gimenes@eq.ufrj.br (M.A.P.G.C.)
[2] Instituto de Química, Universidade Federal do Rio de Janeiro, Rio de Janeiro 21941-909, RJ, Brazil; debora@iq.ufrj.br
\* Correspondence: faustinocarvalho@gmail.com

**Abstract:** A comprehensive study of the thermal deoxygenation of palm residue under sub- and supercritical water conditions using Raney nickel as a heterogeneous catalyst is presented in this paper. Hydrothermal technology was chosen to replace the need for hydrogen as a reactant, as happens, for example, in catalytic hydrotreatment. Several experiments were carried out at different reaction temperatures (350, 370, and 390 °C) and were analyzed with different times of reaction (1, 3.5, and 6 h) and catalyst loads (5, 7.5, 10 wt.%). No hydrogen was introduced in the reactions, but it was produced in situ. The results showed the selectivity of biokerosene ranged from 2% to 67%, and the selectivity of diesel ranged from 5% to 98%. The best result was achieved for 390 °C, 10 wt.% catalyst load, and 3.5 h of reaction, when the selectivities equal to 67% for biokerosene and 98% for diesel were obtained. The Raney nickel catalyst demonstrated a tendency to promote the decarboxylation reaction and/or decarbonylation reaction over the hydrodeoxygenation reaction. Moreover, the fatty acid and glycerol reforming reaction and the water−gas shift reaction were the main reactions for the in situ $H_2$ generation. This study demonstrated that a hydrothermal catalytic process is a promising approach for producing liquid paraffin ($C_{11}-C_{17}$) from palm residue under the conditions of no $H_2$ supply.

**Keywords:** hydrothermal deoxygenation; subcritical water; in-situ $H_2$

## 1. Introduction

The increasing consumption of fossil fuels has led to GHG emissions and global warming, even with their rising prices and apparent depleting resources. Simultaneously, population growth and economic development are spurring increases in demand for new sources of energy and industrial chemical precursors while at the same time producing growing quantities of waste. Consequently, the depletion of fossil fuels, increasing demand for fuels, environmental pollution, and sustainability towards the environment are the movers for biomass as a renewable energy source [1–3].

Second-generation biofuels are currently considered a potential candidate to replace fossil fuels. They reduce greenhouse gas emissions, limit the food versus fuel competition, reduce disposal problems, and there is growing agreement that wastes should be viewed as valuable renewable resources. Waste lipids are attractive for valorization because fatty acid structures have chemical similarities to petroleum-based fuels, and they have high energy density [4].

Waste lipids can also serve as sources for several biochemical and thermochemical pathways to convert this biomass to biofuels. Currently, transesterification processes, gasification (biomass to liquid), Fischer–Tropsch synthesis, hydroprocessing of esters and

fatty acids, catalytic deoxygenation, catalytic hydroprocessing, catalytic cracking, and pyrolysis are most used [5]. These processes, however, have several limitations. For example, the conventional transesterification process (FAME) requires feedstock with high-quality triglycerides and free of moisture, significant inputs of alcohol (methanol or ethanol), and engine modifications if more than 20% FAME is blended with diesel fuel. In contrast, catalytic hydroprocessing requires high costs, high energy requirements, and significant stoichiometric inputs of $H_{2(g)}$ to catalytically reduce oxygenated functional groups present in lipids [1].

As an alternative to conventional refinery processing, the hydrothermal process gained more attraction than other processes since there is a growing interest in valorizing lipid-containing biomass and waste lipid, especially raw materials with high moisture content. The hydrothermal process involves the conversion of feedstock to biofuels and value-added products through the application of high temperature (200–600 °C) and high pressure (5–40 MPa) in the presence of water, aiming at removing oxygen from fatty acids [6]. Moreover, hydrothermal technology gained more attraction than other processes due to its advantages like feedstock flexibility, high energy and resource efficiency of the process, high output product quality, the production of direct substitutes for existing fuels and the presence of efficient heat integration [1,7,8].

Various conditions affect the hydrothermal process, such as temperature, pressure, residence time, and type of catalyst. Temperature is one of the main parameters that affect biofuel selectivity and conversion since it influences the range of the hydrothermal process (fractionation, carbonization, liquefaction, or hydrothermal gasification). Pressure helps maintain water in the liquid state and thus incur savings by avoiding the high energy costs of a liquid–vapor system. Residence time influences product composition and conversion efficiency. The use of catalysts in hydrothermal processes aims to improve the efficiency of the process, mitigating the formation of tar and coal [8].

Currently, there is a preference in the hydrothermal process to use heterogeneous catalysts with a noble metal such as platinum, palladium, and nickel. Nickel has attracted interest as a catalyst for fatty acid deoxygenation since it has a lower price and higher hydrogenation activity than palladium (Pd), platinum (Pt) [6]. Additionally, nickel catalyzes several reactions, such as decarbonylation and decarboxylation reactions, essential to convert saturated fatty acids into hydrocarbons, and side reactions present in the hydrothermal environment, such as hydrogenation, hydrogenolysis, hydrocracking, steam reform, hydrocarbon reform, and methanation. Aqueous-phase reforming of organic acids to generate $CO_2$ and $H_2$ has also been reported over nickel catalysts [9–11].

Among the most used nickel catalysts, there is the commercial Raney nickel catalyst. Raney nickel catalyst was used based on previously published literature [12]. However, most articles concern the use of Raney nickel in reaction systems that use hydrogen as a reactant [13,14]. Notwithstanding, one cannot find articles using Raney nickel catalysts under supercritical conditions and in the absence of hydrogen as a reactant.

In this paper, we explore the potential of the hydrothermal process to generate in-situ $H_2$ from waste lipids and subsequently use it to promote deoxygenation reactions. We investigated the performance of hydrothermal catalytic deoxygenation using palm residue as feedstock. We demonstrated that Raney nickel can fully convert palm oil residue into hydrocarbons in the diesel and biokerosene range through deoxygenation in the sub and supercritical water.

## 2. Results

### 2.1. The Residue of Palm Characterization

The GC-MS results (Table 1) showed that fatty acids were the main products with a total fraction of 78%. The percentage of saturated fatty acids (43%) was higher than unsaturated fatty acids (35%). Hexadecanoic, trans-oleic, and linoleic acid showed higher concentrations and percentages. Glycerides corresponded to 22%, with 18% for Trilaurin, 2% for Monoolein, and 2% for 2-Monopalmitin.

**Table 1.** Quantification of fatty acids and triglycerides present in waste of palm.

| Substance | Concentration (mg g$^{-1}$) | Percentage (%) |
|---|---|---|
| Tetradecanoic acid | 18.25 | 2% |
| Hexadecanoic acid | 374.7 | 34% |
| Linoleic Acid | 110.61 | 10% |
| Trans-Oleic Acid | 258.23 | 23% |
| Oleic acid | 20.91 | 2% |
| Stearic Acid | 74.33 | 7% |
| 2-Monopalmitin | 19.05 | 2% |
| Monoolein | 26.29 | 2% |
| Squalene | Nr | Nr |
| Trilaurin | 137.9 | 18% |
| Trilaurin | 59.75 | |

The presence of trans-oleic acid proves that palm residue is a by-product of the hydrogenated fat industry. It is proven with the palm residue chromatograph, which has a greater area of the trans-oleic acid chromatographic peak than the cis-oleic acid. Consequently, there is a higher concentration of trans-oleic acid in the feedstock, being justified by the hydrogenation reaction of palm oil. When palm oil is hydrogenated, isomerization reactions occur, facilitating a formation of trans- over cis-isomer.

*2.2. Effects of Temperature on Hydrothermal Deoxygenation of the Residue of Palm*

2.2.1. Effect of Temperature during Hydrothermal Treatment of Palm Waste Using Different Loads of Raney Nickel and after 6 h

When 10 wt.% of Raney nickel was used, the conversion (X) of fatty acids and glycerides in palm residue increased from 77% to 97% with the increase in temperature from 350 °C to 370 °C. When the temperature reached 390 °C, the conversion of fatty acids and glycerides reached 100%. Moreover, the selectivity ($S_{Biokerosene}$) of biokerosene went from 10% to 26%, with the increase in temperature from 350 °C to 370 °C, and then it decreased to 14% at 390 °C. The same happened to diesel. The selectivity ($S_{Diesel}$) went up from 20% to 44%, with the temperature ranging from 350 to 370 °C, and then it decreased to 31%, respectively, at 390 °C (Figure 1).

Even with the complete conversion at 390 °C, concentrations of dodecane, tridecane, tetradecane, pentadecane, and heptadecane decreased (Figure S1 on Supporting Information). This decrease in the concentrations of hydrocarbons occurred due to side reactions, such as thermal cracking reactions and steam reforming of hydrocarbon. Both reactions converted hydrocarbons into CO and $H_2$ by steam reforming, and $CH_4$, acetylene ($C_2H_2$), ethylene ($C_2H_4$), ethane ($C_2H_6$), and propane ($C_3H_8$) by thermal cracking [15,16]. Moreover, the aliphatic hydrocarbons and short-chain fatty acids also produce $H_2$, CO, and $CO_2$ through gasification, which is favored above 374 °C [15].

For 7.5 wt.% of Raney nickel, the conversion (X) of fatty acid and glycerides in palm residue increased from 86% at 350 °C to 100% at 370 °C and remained at 100% at 390 °C. The increase in temperature increased the selectivities for biokerosene and diesel (Figure 2). The selectivity of biokerosene ranged from 3% to 14%, when the temperature increased from 350 °C to 370 °C. At 390 °C, the selectivity of biokerosene was 35%. The same occurred for diesel. The selectivity of diesel increased from 5% to 30% when the temperature ranged from 350 °C to 370 °C. At 390 °C, the selectivity of diesel reached 47%. Regarding hydrocarbon concentrations, at 390 °C, the concentration of heptadecane decreased simultaneously with the increase in concentrations of tridecane, tetradecane, and hexadecane and with the production of decane, undecane, dodecane (Figure S2). These results are related to catalytic cracking, which is responsible for shorter hydrocarbons ($C_{10}$–$C_{16}$) [15]. According to Miao et al. [6] and Morgan et al. [17], nickel is a recognized catalyst for breaking C−C of hydrocarbons, and the cracking of hydrocarbons on nickel catalysts becomes more active at higher temperatures. This effect can be advantageously exploited to produce hydrocarbons within the boiling point range of aviation-grade fuels.

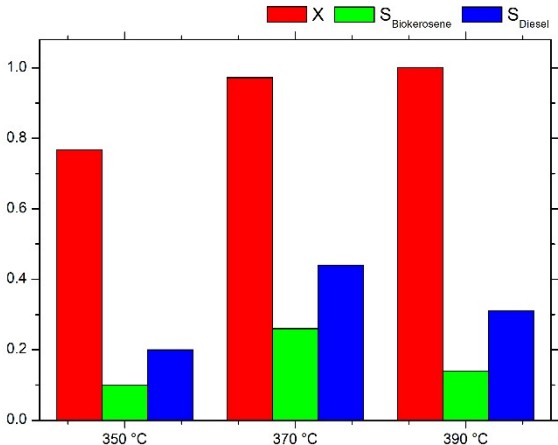

**Figure 1.** Effect of temperature on the conversion (X) of palm residue (%), selectivities to biokerosene (%), and selectivities to diesel (%); Experimental conditions: 6 h, 10 wt.% catalyst, biomass:water equals 1:2).

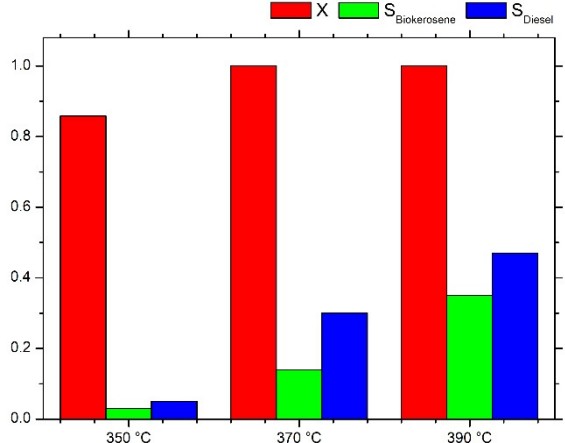

**Figure 2.** Effect of temperature on the conversion of palm residue (%), selectivities to biokerosene (%), and selectivities to diesel (%). Experimental conditions: 6 h, 7.5 wt.% Raney nickel, biomass:water equals 1:2.

Conversely, for 5 wt.% in Raney nickel, the conversion (X) was 100% at 350 °C, 99% at 370 °C, and 100% at 390 °C. The selectivity of biokerosene went up from 21% to 35% with the increase in temperature from 350 to 370 °C, and then decreased to 15% at 390 °C. The same happened to the selectivity of diesel. It increased from 42% to 52% when temperature ranged from 350 °C to 370 °C, then decreased to 29% at 390 °C (Figure 3).

When the temperature reached 370 °C using 5 wt.% Raney nickel, concentrations of pentadecane and heptadecane decreased. These, however, were not enough to affect the selectivity for biokerosene and diesel. Consequently, it is possible to conclude that deoxygenation reactions were favored up to 370 °C. At 390 °C, however, undecane and dodecane disappeared, and the concentrations of tridecane, tetradecane, pentadecane, hexadecane, and heptadecane decreased, affecting the selectivity for biokerosene and diesel (Figure S3). These results were related to the presence of thermal cracking reactions and steam reforming of hydrocarbons that generate $H_2$, CO, and $CO_2$, and $CH_4$, $C_2H_2$, $C_2H_4$, $C_2H_6$, and $C_3H_8$, respectively, and gasification that produces $H_2$, CO, and $CO_2$ [18,19]. Moreover, cracking reactions and steam reforming were favored at high temperatures [20].

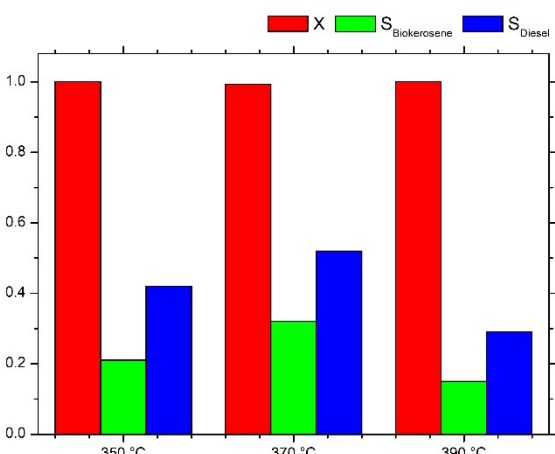

**Figure 3.** Effect of temperature on the conversion of palm residue selectivities to biokerosene (%), and selectivities to diesel (%). Experimental conditions: 6 h, 5 wt.% Raney nickel, biomass:water equals 1:2.

2.2.2. Effect of Temperature during Hydrothermal Treatment of Palm Residue Using Different Loads of Raney Nickel and after 3.5 h

When 10 wt.% of Raney nickel was used, the conversion (X) of palm wastes increased slightly from 87% to 100%, when temperature increased from 350 °C to 370 °C and remained at 100% at 390 °C. The selectivity of biokerosene and diesel increased continuously as the reaction temperature increased. The selectivity of biokerosene ranged from 18% to 22% when temperature increased from 350 °C to 370 °C and increased to 67% at 390 °C. For diesel, the selectivity increased from 32% to 44% when the temperature ranged from 350 °C to 370 °C and then increased to 98% at 390 °C (Figure 4). The preference of deoxygenation reactions can justify these results, which explain the increase in the concentrations of decane, undecane, dodecane, tridecane, tetradecane, pentadecane, hexadecane, and heptadecane (Figure S4).

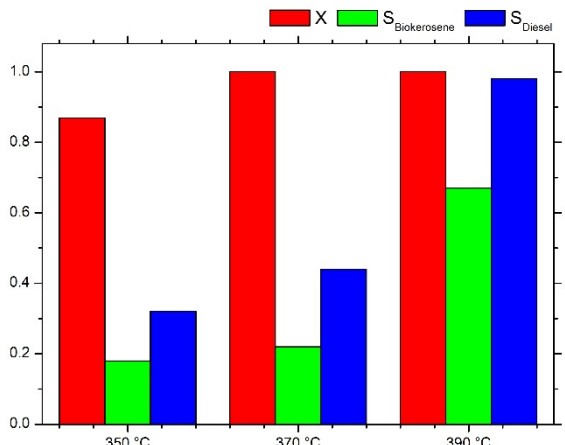

**Figure 4.** Effect of temperature on the conversion of palm residue selectivities to biokerosene (%), and selectivities to diesel. Experimental conditions: 3.5 h, 10 wt.% Raney nickel, biomass:water equals 1:2.

The conversion (X) of palm residue using 7.5 wt.% of Raney nickel decreased from 96% to 90% at 350 °C and 370 °C, respectively, and reduced to 82% at 390 °C (Figure 5). One explanation for these low conversions is the deactivation of nickel and subsequent loss of activity due to the oxidation of nickel in the aqueous phase and forming carbon deposits that prevent access of the reactant molecules to active sites. The insufficient concentration of $H_2$ produced in situ also affects the reduction in conversion [6,16]. Increasing concentrations of tetradecanoic, hexadecanoic, and stearic acids found at 370 °C and 390 °C in GC-MS analysis justified these scenarios.

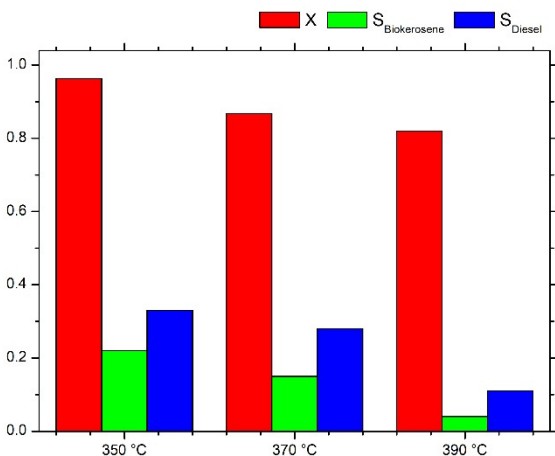

**Figure 5.** Effect of temperature on the conversion of palm residue selectivities to biokerosene (%), and selectivities to diesel (%). Experimental conditions: 3.5 h, 7.5 wt.% Raney nickel, biomass:water equals 1:2.

Selectivities of biokerosene and diesel decreased, respectively, from 22% to 15% and from 33% to 28%, as the temperature increased from 350 °C to 370 °C. At 390 °C, selectivities decreased again to 4% and 3% for biokerosene and 11% and 9% for diesel. This discrepancy could result from the production of gaseous by-products such as $CH_4$ and $C_2H_6$ caused by the thermal cracking or steam reforming of hydrocarbons [15]. Moreover, the hydrothermal process at 390 °C favors gasification reactions that convert fatty acids into fuel gas ($H_2$, CO, $CO_2$) [16]. Consequently, the rate of cracking reaction, steam reforming of hydrocarbons, and gasification were more significant than the rate of hydrocarbon formation. Figure S5 shows the concentrations of hydrocarbons, and conversion and selectivity values.

Using 5 wt.% of Raney nickel, the conversion (X) increased from 81% to 98%, when the temperature ranged from 350 °C to 370°C and decreased to 86% at 390 °C. The preference of secondary reactions favored by high temperatures, by the loss of nickel activity due to oxidation and carbon deposition at the catalyst's active site, and by the insufficient concentration of $H_2$ can justify the decrease in conversion at 390 °C [6,20]. These hypotheses for decreasing conversion were proven by the higher concentration of hexadecanoic and stearic acids identified at 390 °C.

Selectivities of biokerosene increased from 3% to 17%, at 350 °C and 370 °C, respectively. Then the selectivity of biokerosene decreased to 9% at 390 °C. The same happened to diesel. The selectivity increased from 6% to 37% when the temperature increased from 350 °C to 370 °C, and decreased to 19% at 390 °C (Figure 6).

Comparing 370 °C to 350 °C, the concentrations of tetradecanoic, hexadecanoic, and stearic acids decreased. The concentration of hydrocarbons increased from 55.28 mg g$^{-1}$ to 397.86 mg g$^{-1}$, resulting from the appearance of tridecane, tetradecane, and hexadecane and increased concentrations of pentadecane and heptadecane (Figure S6). Thus, it is possible to conclude that a temperature of 370 °C benefited reactions of deoxygenation. At 390 °C, the concentration of hydrocarbons decreased again to 176.82 mg g$^{-1}$, which proved the preference for thermal cracking, steam reforming, and gasification over $C_{13}$ to $C_{17}$.

### 2.2.3. Effect of Temperature during Hydrothermal Treatment of Palm Residue Using Different Loads of Raney Nickel and after 1 h

The conversion (X) of palm residue using 10 wt.% of Raney nickel increased from 86% to 87%, at 350 °C and 370 °C, respectively. At 390 °C, the conversion of palm residue decreased to 61%. At 390 °C, hydrogenolysis and thermal cracking occurred, and were confirmed by the identification of new fatty acids such as octanoic, decanoic, undecanoic, dodecanoic, tridecanoic, pentadecanoic acids. This, however, was not enough to increase the conversion. The low conversion at 390 °C can be explained by the deactivation of nickel catalyst and loss of activity caused by CO poisoning or nickel oxidation in the

aqueous phase [6]. This explanation is confirmed by detecting oleic and stearic acids, which corresponded to 60% of the sample.

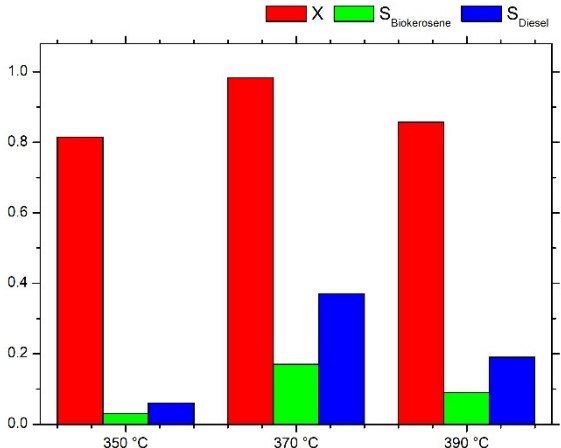

**Figure 6.** Effect of temperature on the conversion of palm residue (%),selectivities to biokerosene (%), and selectivities to diesel (%). Experimental conditions: 3.5 h, 5 wt.% Raney nickel, biomass:water equals 1:2.

The selectivity of biokerosene increased from 10% to 11% when the temperature ranged from 350 °C to 370 °C and reached 24% at 390 °C. For diesel, the selectivity decreased from 21% to 19% when the temperature increased from 350 °C to 370 °C and increased to 48% at 390 °C (Figure 7).

Even with low conversion at 390 °C, the selectivities for biokerosene and diesel were higher than the values obtained at 350 °C and 370 °C. This demonstrated that the temperature of 390 °C favored the deoxygenation reactions on the fatty acids considered intermediate and formed from hydrogenation, thermal cracking, and hydrogenolysis reactions. Concentrations of heptadecane, hexadecane, pentadecane, and tetradecane increased when the temperature ranged from 370 °C to 390 °C (Figure S7).

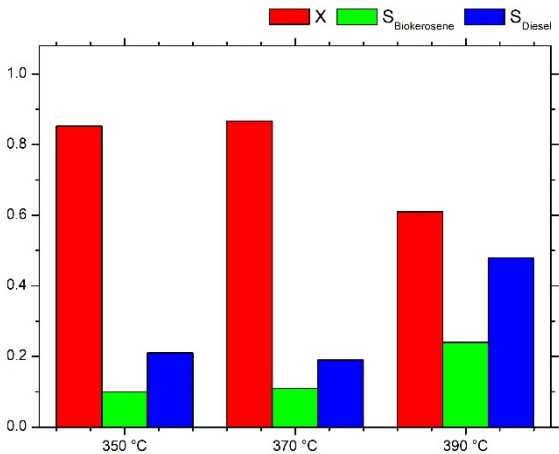

**Figure 7.** Effect of temperature on the conversion of palm residue (%), selectivities to biokerosene (%), and selectivities to diesel (%).. Experimental conditions: 1 h, 10 wt.% Raney nickel, biomass:water equals 1:2.

For 5 wt.% of Raney nickel, the conversion (X) of palm residue decreased continuously as the reaction temperature increased. The conversion dropped from 89% to 84%, increasing temperature from 350 to 370 °C and again reduced to 77% at 390 °C. The selectivity of biokerosene decreased from 8% at 350 °C to 3% at 370 °C and then increased to 7% at 390 °C. The same happened to the selectivity of diesel (Figure 8). Concerning hydrocarbons, 350 °C produced the highest concentration of hydrocarbons with 153.55 mg g$^{-1}$, and 370 °C

formed the lowest concentration of hydrocarbons with 63.97 mg g$^{-1}$. Figure S8 shows all hydrocarbons produced at 350 °C, 370 °C, and 390 °C.

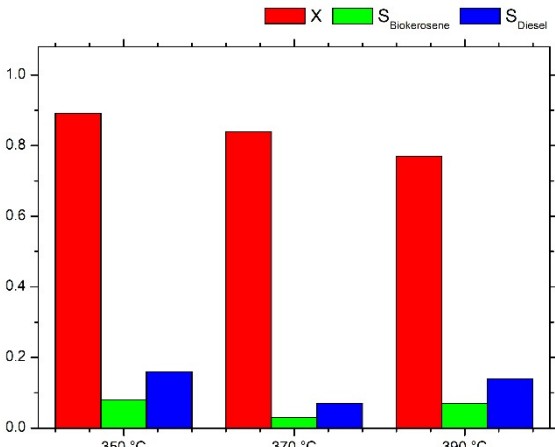

**Figure 8.** Effect of temperature on the conversion of palm residue (%), selectivities to biokerosene (%), and selectivities to diesel (%).. Experimental conditions: 1 h, 5 wt.% Raney nickel, biomass:water equals 1:2.

### 2.3. Reactions Scheme in the Hydrothermal Process of Palm Residue

The analysis of residue of palm using GC-MS indicated that it contains only fatty acids and glycerides. Glycerides identified were 2-Monopalmitin, Monoolein, and Trilaurin, a dodecanoic acid triglyceride (1,2,3-Tridodecanoylglycerol). Fatty acids detected were tetradecanoic, hexadecanoic, stearic, linoleic, oleic acids (trans and cis isomers).

A qualitative analysis by GC-MS of the products obtained after hydrothermal experiments detected 23 analytes (14 fatty acids and nine hydrocarbons). This allowed the identification of different reactions in the hydrothermal process. Likewise, the coexistence of the liquid and gaseous phases of the water and the presence of triglycerides, monoglycerides, and saturated and unsaturated fatty acids also triggered the simultaneous occurrence of different reactions in the hydrothermal environment. The reactions identified are the steam reforming of glycerol, steam reforming of fatty acids, steam reforming of hydrocarbons, water–gas shift reaction, hydrolysis, hydrogenation, hydrogenolysis, thermal cracking, catalytic cracking, and methanation. Most of the hydrothermal process, with triglycerides and fatty acids, preferred to use heterogeneous catalysts with a noble metal such as platinum, palladium, Nickel. Nickel, compared to palladium (Pd) and platinum (Pt), has a lower price and high activity for hydrogenation and methanation. Additionally, it has other important characteristics such as its high reactivity and affinity to $\pi$ systems and the possibility of access to different oxidation states [19,21]. Moreover, Nickel catalyzes several reactions such as decarbonylation, de-carboxylation, hydrogenation, hydrogenolysis, hydrocracking, steam reform, hydrocarbon reform, and methanation [17,20,22].

GC-MS analyses of the samples after the hydrothermal process did not identify 2-Monopalmitin, Monoolein, and Trilaurin. Hence, they must have been transformed through hydrolysis reactions. The hydrolysis reaction is a reversible reaction of the first order. In the first step, the triglyceride is hydrolyzed to diglyceride; in the second stage, the diglyceride is hydrolyzed to monoglyceride; and in the third stage, the monoglyceride is hydrolyzed to fatty acid and glycerol. At each stage, there is the formation of a fatty acid [22].

According to Vardon et al. [18], free fatty acids generated from lipid hydrolysis can contain varying ratios of saturated, unsaturated, and polyunsaturated fatty acids, depending on the feedstock origin. For Trilaurin (Figure S9), e.g., there is the formation of three molecules of dodecanoic acid and one molecule of glycerol. For 2-Monopalmitin (Figure S10), a hexadecanoic acid molecule and a glycerol molecule were formed; for monoolein (Figure S11), the hydrolysis produced one molecule of oleic acid and glycerol (Figure 9).

**Figure 9.** Lipid hydrolysis representation.

The linoleic acid was not detected after hydrothermal experiments. Consequently, linoleic acid was hydrogenated, producing oleic acid. In the same way, oleic acid was not identified too, and it was also hydrogenated to stearic acid. The disappearance of C=C to C-C indicated the Raney nickel catalyst is a hydrogenation catalyst. The overall amount of the formed hydrogen was sufficient to complete the hydrogenation of linoleic and oleic acids into stearic acid. Figure 10 shows the hydrogenation from linoleic acid to stearic acid.

**Figure 10.** Hydrogenation of linoleic acid.

The increase in the percentage of stearic acid in some samples after the hydrothermal reaction proved the existence of hydrogenation reactions. In the residue of palm, the percentage of stearic acid was 7%. The maximum content of stearic acid reached 34% at 350 °C, 7.5 wt.% of Raney nickel and 6 h; 350 °C, 7.5 wt.% of Raney nickel and 1 h; and at 370 °C, 5 wt.% of Raney nickel and 1 h. Moreover, the hydrogen concentration also affects the catalyst's performance in decarboxylation [19].

The hydrogenation reaction happens with the presence of hydrogen. All hydrothermal experiments generated hydrogen in situ. The formation of hydrogen can occur due to aqueous-phase reforming of glycerol, steam reforming of fatty acids, and water–gas shift reaction [16,17]. Most palm residue compounds are free fatty acids, and 35% of the sample corresponds to unsaturated fatty acids. Thus, the hydrogenation reaction is the limiting step of the hydrothermal process. If the hydrogen concentration is insufficient, this will limit the hydrogenation. Consequently, the total conversion of unsaturated to saturated fatty acids will not occur, and this will decrease the selectivity for deoxygenation reactions and the formation of hydrocarbons. Moreover, the low concentration of $H_2$ also causes incomplete deoxygenation of palm residue.

The presence of 22% of glycerides (trilaurin, 2-monopalmitin, and monoolein) in the palm residue improves the conversion of unsaturated fatty acids through the aqueous phase reforming of glycerol that transforms the glycerol produced into $H_2$ [6,19]. Therefore, the addition of glycerol is not necessary since the hydrolysis of triglycerides produces glycerol molecules. Additionally, CO generated by aqueous-phase reforming of glycerol is consumed to produce additional $H_2$ via the water–gas shift reaction [18]. Alwan, Salley and Ng [19] proved that the addition of glycerol to hydrothermal decarboxylation reagents improves the conversion of free unsaturated fatty acids. At the same time, Hollak et al. [23] demonstrated the increase in hydrothermal deoxygenation activity with the presence of excess glycerol. The steam reforming of fatty acids can not be ignored either since the palm residue corresponds to 79% of the sample. According to Miao et al. [6], steam reforming of fatty acid facilitates the formation of $H_2$ in situ.

The concentrations of hexadecanoic and tetradecanoic acids decreased. This decrease occurred through decarboxylation/decarbonylation reactions to produce pentadecane and tridecane; hydrodeoxygenation to form hexadecane and tetradecane; hydrogenolysis reactions to generate shorter fatty acids such as pentadecanoic and tridecanoic acids; hydrogenolysis reactions combined with decarboxylation/decarbonylation reactions to form hydrocarbons with shorter carbon chains; decarboxylation/decarbonylation reactions combined with catalytic cracking to form hydrocarbons with shorter carbon chains ($C_{10}$–$C_{16}$); thermal cracking reactions to produce $CH_4$, $C_2H_2$, $C_2H_4$, $C_2H_6$, $C_3H_8$; or steam reforming to form $H_2$, CO e $CO_2$ [16,19,20].

The deoxygenation hydrothermal of palm residue generated other fatty acids such as hexanoic, heptanoic, octanoic, nonanoic, decanoic, undecanoic, dodecanoic, tridecanoic, pentadecanoic, and heptadecanoic acids. These fatty acids, except dodecanoic acid generated by hydrolysis of Trilaurin, confirmed the presence of successive hydrogenolysis and thermal cracking. The hydrogenolysis cleaves the C-C bond and produces fatty acids with lower molecular weights and $CH_4$ as a side product [6,20]. Hexanoic acid was the smallest detectable fatty acid after the hydrothermal process.

The hydrocarbons detected were decane, undecane, dodecane, tridecane, tetradecane, pentadecane, hexadecane, and heptadecane. Additionally, heptadecane and pentadecane were observed as the major hydrocarbon products. Deoxygenation reactions (decarboxylation, decarbonylation, and hydrodeoxygenation reactions) formed all hydrocarbons. While the decarboxylation reaction does not require the presence of $H_2$ and forms $C_{(n-1)}$ hydrocarbons and $CO_2$, the decarbonylation reaction requires an $H_2$ molecule and produces $C_{(n-1)}$ hydrocarbons, CO and $H_2O$. The hydrodeoxygenation reaction needs hydrogen and forms hydrocarbons with a Cn carbon chain + $nH_2O$ [6].

All experiments showed a higher concentration of pentadecane and heptadecane than hexadecane and octadecane (octadecane was not produced); so, the Raney nickel catalyst promotes decarboxylation/decarbonylation rather than hydrodeoxygenation [4]. Table 2 shows the hydrocarbon concentrations after the hydrothermal reaction.

**Table 2.** Concentration (mg g$^{-1}$) of hydrocarbons ($C_{11}$–$C_{17}$) after the hydrothermal process of palm residue.

| Temperature (°C) | Catalyst (wt.%) | Time (h) | $C_{10}$ | $C_{11}$ | $C_{12}$ | $C_{13}$ | $C_{14}$ | $C_{15}$ | $C_{16}$ | $C_{17}$ |
|---|---|---|---|---|---|---|---|---|---|---|
| 390 | 10 | 6 | 0.00 | 0.00 | 0.00 | 5.47 | 21.65 | 128.32 | 41.17 | 144.32 |
| 390 | 10 | 3.5 | 32.33 | 75.04 | 104.12 | 130.64 | 138.84 | 256.71 | 104.19 | 237.03 |
| 390 | 10 | 1 | 0.00 | 0.00 | 0.00 | 12.16 | 23.44 | 125.35 | 23.49 | 135.83 |
| 390 | 7.5 | 6 | 15.00 | 39.38 | 55.96 | 70.40 | 71.70 | 135.52 | 37.36 | 92.38 |
| 390 | 7.5 | 3.5 | 0.00 | 0.00 | 0.00 | 0.00 | 6.40 | 25.06 | 8.56 | 61.23 |
| 390 | 7.5 | 1 | 0.00 | 0.00 | 53.90 | 71.50 | 76.79 | 159.53 | 57.19 | 154.78 |
| 390 | 5 | 6 | 0.00 | 0.00 | 0.00 | 7.45 | 26.32 | 130.70 | 31.96 | 119.32 |
| 390 | 5 | 3.5 | 0.00 | 0.00 | 0.00 | 0.00 | 0.00 | 85.77 | 0.00 | 91.05 |
| 390 | 5 | 1 | 0.00 | 0.00 | 0.00 | 4.53 | 7.93 | 46.65 | 8.21 | 52.07 |
| 370 | 10 | 6 | 0.00 | 0.00 | 23.78 | 40.32 | 47.03 | 162.48 | 35.71 | 157.09 |
| 370 | 10 | 3.5 | 0.00 | 0.00 | 0.00 | 13.69 | 33.00 | 199.39 | 35.79 | 202.19 |
| 370 | 10 | 1 | 0.00 | 0.00 | 0.00 | 14.78 | 17.57 | 73.00 | 0.00 | 75.81 |
| 370 | 7.5 | 6 | 0.00 | 0.00 | 0.00 | 3.05 | 18.67 | 134.93 | 30.27 | 148.03 |
| 370 | 7.5 | 3.5 | 6.7 | 17.57 | 25.13 | 36.14 | 39.34 | 212.05 | 28.22 | 200.40 |
| 370 | 5 | 6 | 0.00 | 19.59 | 33.42 | 53.11 | 60.70 | 181.66 | 44.35 | 173.96 |
| 370 | 5 | 3.5 | 0.00 | 0.00 | 0.00 | 8.75 | 24.07 | 154.31 | 27.77 | 158.90 |
| 370 | 5 | 1 | 0.00 | 0.00 | 0.00 | 0.00 | 0.00 | 29.63 | 0.00 | 34.34 |
| 350 | 10 | 6 | 0.00 | 0.00 | 0.00 | 5.42 | 8.60 | 70.58 | 9.59 | 85.97 |
| 350 | 10 | 3.5 | 0.00 | 0.00 | 0.00 | 26.77 | 32.32 | 109.19 | 22.86 | 111.77 |
| 350 | 10 | 1 | 0.00 | 0.00 | 0.00 | 4.28 | 11.03 | 80.79 | 12.80 | 91.71 |
| 350 | 7.5 | 6 | 0.00 | 0.00 | 0.00 | 0.00 | 0.00 | 23.75 | 0.00 | 25.05 |
| 350 | 7.5 | 3.5 | 0.00 | 0.00 | 4.03 | 19.57 | 33.16 | 177.12 | 28.49 | 83.24 |
| 350 | 7.5 | 1 | 0.00 | 0.00 | 0.00 | 0.00 | 0.00 | 21.14 | 0.00 | 24.19 |
| 350 | 5 | 6 | 0.00 | 0.00 | 0.00 | 14.97 | 32.27 | 185.72 | 32.38 | 194.38 |
| 350 | 5 | 3.5 | 0.00 | 0.00 | 0.00 | 0.00 | 0.00 | 26.72 | 0.00 | 28.57 |
| 350 | 5 | 1 | 0.00 | 1.42 | 3.51 | 6.90 | 9.14 | 60.88 | 7.98 | 63.72 |

There are two possible pathways to the formation of short-chain paraffins ($C_{10}$–$C_{13}$). One is through hydrogenolysis, with cleavage of C-C bond and the production of fatty acids with lower molecular weights, and the subsequent decarboxylation/decarbonylation of fatty acid formed. The other is the decarboxylation/decarbonylation reaction combined with catalytic cracking. The presence of fatty acids produced by hydrogenolysis and hydrocarbons via deoxygenation reactions (decarboxylation, decarbonylation, and hydrodeoxygenation) demonstrated the preference of the first pathway [6].

According to the results, a viable hydrothermal mechanism for the residue of palm can be proposed. Firstly, three molecules of dodecanoic acid and one molecule of glycerol were formed through hydrolysis. Simultaneously, 2-Monopalmitin and Monoolein are hydrolyzed to hexadecanoic acid and linoleic acid (one molecule each), respectively, and glycerol. Glycerol liberated by hydrolysis may undergo aqueous-phase reforming to produce $H_2$ or catalytically decomposes to generate CO. The carbon monoxide will be successively consumed by water–gas shift reaction for additional $H_2$ production [18,19]. At the same time, there is steam reforming of fatty acid to form $H_2$.

Accompanied by in situ $H_2$ production, thermal or catalytic cracking and successive hydrogenolysis generate shorter chain fatty acids [20]. According to Nanda et al. [16], each feedstock has an ideal thermal cracking range. For palm oil, temperatures from 350 °C to 400 °C favor the thermal cracking of fatty acids, and all experiments were carried out in that range. As the temperature increases, the more active the nickel becomes, which favors catalytic cracking reactions [6,19]. At the same time, the long residence time favors better cracking reactions [24].

Unsaturated fatty acids suffer hydrogenation reactions to generate saturated fatty acids. Moreover, there is the production of new fatty acids. The detection of hexanoic, heptanoic, octanoic, nonanoic, decanoic, undecanoic, dodecanoic, tridecanoic, pentadecanoic, and heptadecanoic acids after the hydrothermal experiment confirmed that at first happen the hydrogenolysis, thermal cracking of fatty acids, and following occur deoxygenation reactions (decarboxylation, decarbonylation or hydrodeoxygenation reactions). This agrees with the theory of Zhang et al. [25].

Deoxygenation reactions produce hydrocarbons. As the Raney nickel catalyst promotes decarboxylation and decarbonylation reactions over hydrodeoxygenation reaction, aliphatic hydrocarbons will be produced with a carbon number equal to $C_{n-1}$ and with the loss of one molecule of $CO_2$ (decarboxylation) or CO (decarbonylation). The $CO_2$ and CO can be used in methanation reactions to form methane.

Fatty acids and hydrocarbons may undergo other reactions. In this sense, steam reforming to produce $H_2$, CO, and $CO_2$, thermal cracking to form $CH_4$, acetylene, ethene, ethane, and propane, and catalytic cracking to generate light hydrocarbons ($C_{10}$–$C_{16}$) [15,19]. Moreover, there are gasification and methanation. The gasification converts hydrocarbons and short-chain fatty acids into $H_2$, CO, and $CO_2$. At the same time, methanation is favored between 100–400 °C and high pressure. In fact, hydrogen may react with CO or $CO_2$ to produce $CH_4$, decreasing the hydrogen concentration in situ. Both affect the selectivity of biokerosene and diesel [6,15,16,26]. Figures 11–14 show the proposed reactions in the hydrothermal process of Trilaurin, 2-Monopalmitin, and monoolein, respectively.

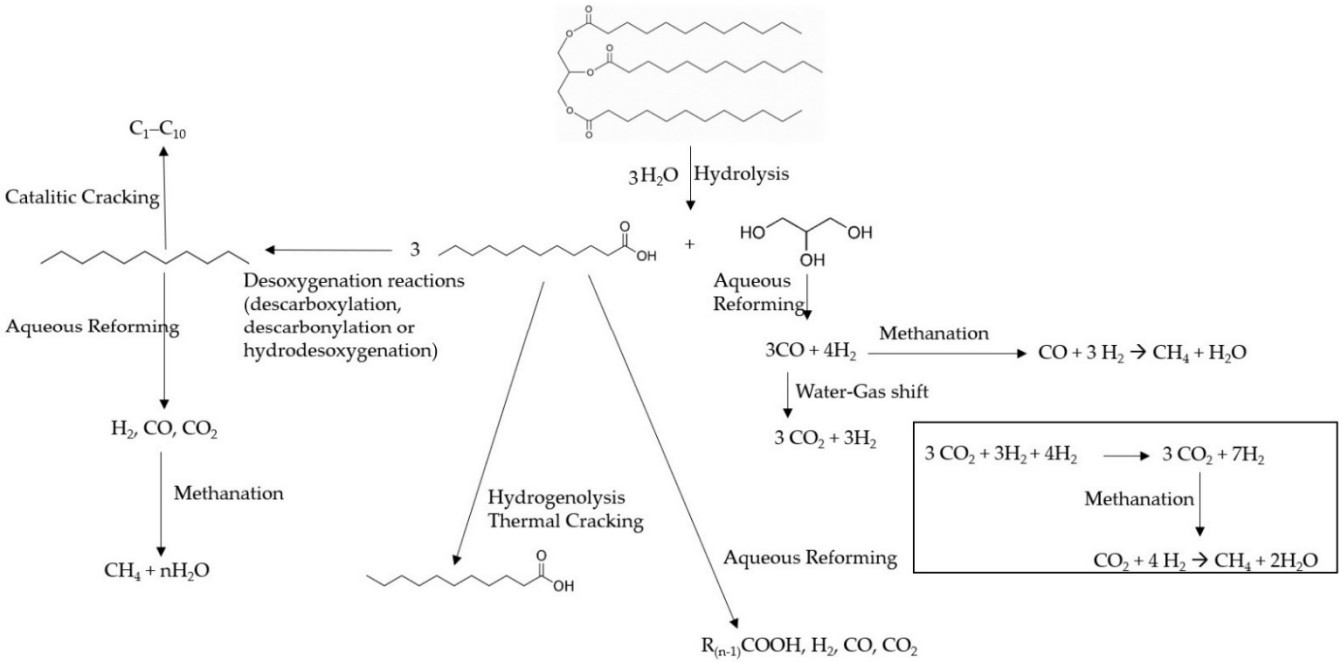

**Figure 11.** Proposed reaction scheme for hydrothermal processing of the Trilaurin into hydrocarbons.

**Figure 12.** Deoxygenation reactions (decarboxylation, decarbonylation, and hydrodeoxygenation) of dodecanoic acid.

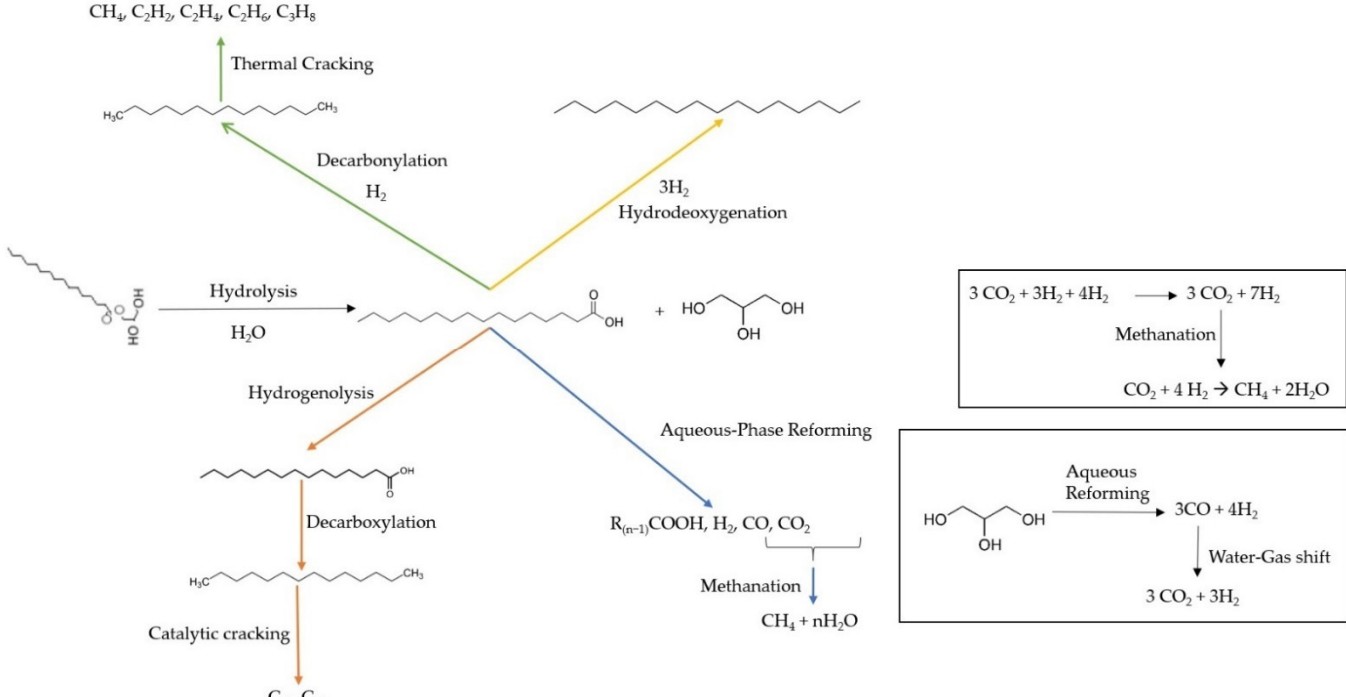

**Figure 13.** Proposed reaction scheme for hydrothermal processing of the 2-monopalmitin into hydrocarbons.

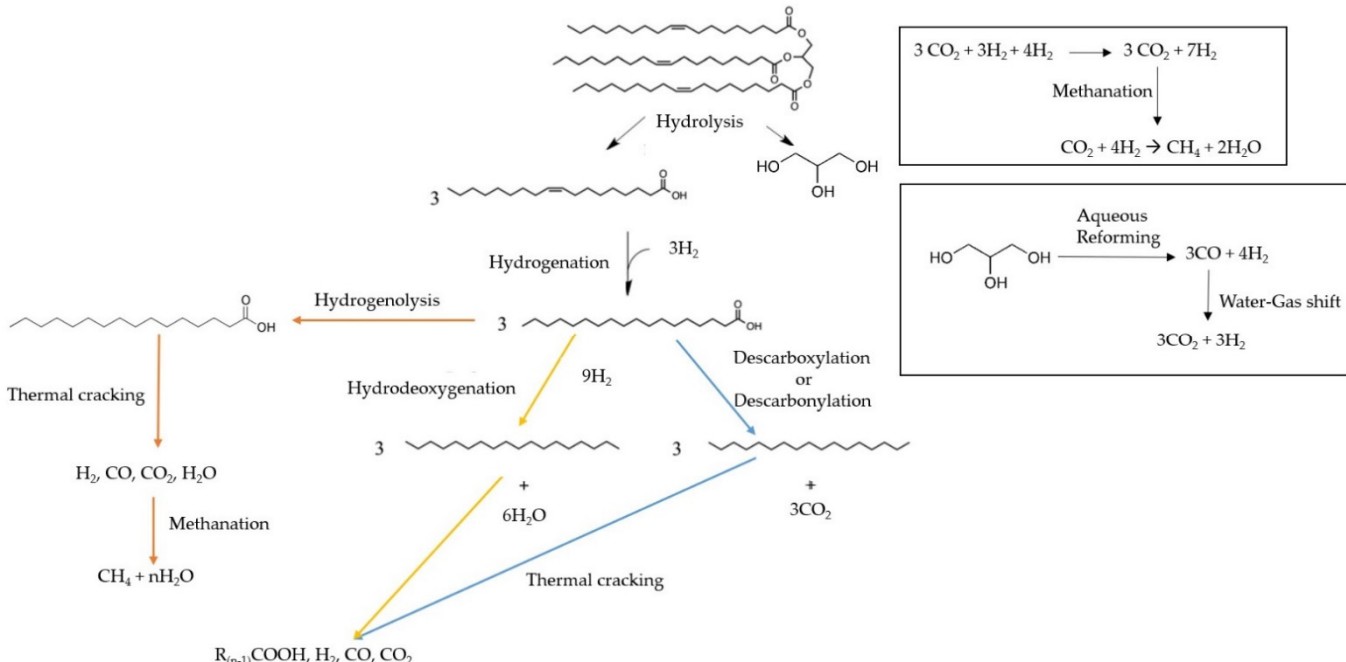

**Figure 14.** Proposed reaction scheme for hydrothermal processing of the monoolein into hydrocarbon.

## 3. Materials and Methods

### 3.1. Materials

Raney nickel was obtained from Laboratório de Estudos em Alcoolquímica e Catálise (LEACAT). The palm residue was purchased from the company M. Dias Branco S.A (Fortaleza, CE, Brazil). The silica was purchased from the Silicycle brand (Silica Siliaflash G60, 60–200 μM, 60 Å, MESH 70–230 mesh). Deuterated decanoic acid (≥98%), n-hexadecane-D34 (≥98%) and MSTFA (*N*-Methyl-*N*-(trimethylsilyl) trifluoroacetamide)

(≥98.5%), and dichloromethane (≥99.9%) were purchased from Sigma Aldrich—St. Louis, MO, USA.

### 3.2. Reaction Procedure

Catalytic hydrothermal reactions were conducted in an unstirred mini-reactor assembled from 3/8-inch 316L stainless steel, 15 cm long and with a volume of 1.7 mL, sealed with a cap on each end.

In each experiment, 400 μL of palm residue, 800 μL of distilled water, and a mass percentage loading of Raney nickel were loaded into the reactor. Biomass to water ratio of 1:2 was employed. The reactor was filled to 50% of the reactor tube volume at room temperature. All experiments were conducted to see the impacts of temperature (350, 370, and 390 °C), reaction time (1, 3.5, and 6 h), and the mass percentage loading of Raney nickel (5 wt.%, 7.5 wt.%, and 10 wt.%). The pressures were autogenous and were generated by temperature during the hydrothermal process.

Subsequently, the sealed reactor was placed inside a small furnace, with a fixed temperature for each experiment. After the reaction, the reactors were submerged into a cooling bath to reduce the temperature to 20 °C. After 10 min to condense the volatile organic products, the reactor top was then removed to empty the reactor of its liquid and solid contents. The products were transferred to volumetric flasks, and the reactors were rinsed with repeated dichloromethane washes until the total volume collected was 5 mL. Afterward, the products were introduced in column chromatography to remove all the catalysts and the water present in the organic phase.

### 3.3. Analysis Method

The identification of the products was analyzed by gas chromatography–mass spectrometry (GC-MS) using an Agilent 6890 Series GC system (Agilent Technologies, Santa Clara, CA, USA). Derivatized sample (1 μL) was injected into an HP-5ms capillary column (Agilent Technologies, Palo Alto, CA, USA) with (5%-phenyl)-methylpolysiloxane phase (30 m × 0.25 mm i.d. × 0.25 μm) in a split mode (ratio 5:1). The injector temperature was set at 325 °C. Helium (purity of 99.9%) was used as the carrier gas at a constant flow rate of 2.0 mL min$^{-1}$. The initial column temperature was 40 °C, which was then increased to 240 °C at 6 °C min$^{-1}$, from 240 °C to 320 °C at 20 °C min$^{-1}$, and at 320 °C for 5 min.

The samples were derivatized to increase the volatility of the components of the fatty acid. These were prepared by adding 1 mL dichloromethane, 1 mL (with 54.5 μg mL$^{-1}$) of deuterated decanoic acid as the internal standard and 1 mL *N*-Methyl-*N*-(trimethylsilyl) trifluoroacetamide to ∼1 mg of the sample and heating for 30 min at 60 °C. After reacting, the solution was allowed to cool to room temperature. Additionally, before the injection of samples into the gas chromatograph, 1 mL n-hexadecane-D34 (with 54.3 5 μg mL$^{-1}$) was added as the internal standard to identify and quantify the liquid hydrocarbon products. Only peaks with similar mass spectra higher than 70% were tentatively identified by comparing the mass spectra of the detected compounds with those found in the NIST (Mass Spectral version 2.0) commercial library.

Conversion (Equation (1)) and selectivity of the product (biokerosene and renewable diesel) (Equation (2)) were calculated using the following equations:

$$X(\%) = \frac{\text{Initial concentration} - \text{Final concentration}}{\text{Initial concetration}} \times 100 \tag{1}$$

$$S(\%) = \frac{\Delta(\text{Concentration of X})}{-\Delta(\text{Concentration of coconut oil}) \times \text{stoichiometric coefficient of X}} \tag{2}$$

## 4. Conclusions

The hydrothermal process of the residue of palm using different loading of Raney nickel catalyst produced hydrocarbons at different temperatures and reaction times.

The most significant fatty acid identified in hydrothermal experiments from the residue of palm was stearic acid. The Raney nickel catalyst demonstrated a preference for decarboxylation or decarbonylation reactions over the hydrodeoxygenation reaction. Raney nickel was also responsible for the reactions of hydrogenolysis, catalytic cracking, and steam reforming. Consequently, the formation of new fatty acids proved Raney nickel's preference in catalyzing hydrogenolysis reactions before catalyzing deoxygenation reactions.

The selectivity of biokerosene ranged from 2% to 67%, and diesel from 5% to 98%. The best result was at 390 °C, 10 wt.% of Raney nickel, and 3.5 h with selectivity equal to 67% for biokerosene and 98% for diesel. The low selectivity for biokerosene is related to favoring secondary reactions resulting from severe conditions of temperature and pressure, and reactions favored by the Raney nickel catalyst. There is also the favoring of gasification reactions when operating at temperatures above 374 °C. Moreover, the deactivation of the nickel catalyst also influences the low selectivity.

Another observation is related to the high conversion values not being directly linked to deoxygenation reactions. Palm residue has different compounds that made the reactions complex in the hydrothermal medium. At the same time, the coexistence of the liquid and gaseous phases of water in the subcritical and supercritical water is also responsible for numerous side reactions.

Raney nickel showed great potential to hydrolyze triglycerides, generate $H_2$ in situ from glycerol, hydrogenate oleic acid, and linoleic acid to form stearic acid and produce hydrocarbons. However, additional hydrothermal decarboxylation is necessary to increase the selectivity of hydrocarbons (diesel and biokerosene). Thus, Raney nickel catalysts may provide an economically viable process for the hydrothermal decarboxylation of fatty acids and triglycerides without the need for additional $H_2$.

Finally, the obtained results conclude that the hydrothermal process with palm residue can produce paraffins present in biokerosene and diesel and is an excellent tool for replacement some processes such as Fischer–Tropsch synthesis, hydroprocessing of esters and fatty acids, catalytic deoxygenation, catalytic cracking, and pyrolysis.

**Supplementary Materials:** The following are available online at https://www.mdpi.com/article/10.3390/catal11080995/s1, Figure S1. Hydrocarbon quantification by GC-MS for the reaction carried out at 10 wt.% of catalyst, 6 h, and biomass:water ratio of 1:2; Figure S2. Hydrocarbon quantification by GC-MS for the reaction carried out at 7.5 wt.% of catalyst, 6 h, and biomass:water ratio of 1:2; Figure S3. Hydrocarbon quantification by GC-MS for the reaction carried out at 5 wt.% of catalyst, 6 h, and biomass:water ratio of 1:2; Figure S4. Hydrocarbon quantification by GC-MS for the reaction carried out at 10 wt.% of catalyst, 3.5 h, and biomass:water ratio of 1:2; Figure S5. Hydrocarbon quantification by GC-MS for the reaction carried out at 7.5 wt.% of catalyst, 3.5 h, and biomass:water ratio of 1:2; Figure S6. Hydrocarbon quantification by GC-MS for the reaction carried out at 5 wt.% of catalyst, 3.5 h, and biomass:water ratio of 1:2; Figure S7. Hydrocarbon quantification by GC-MS for the reaction carried out at 7.5 wt.% of catalyst, 1 h, and biomass:water ratio of 1:2; Figure S8. Hydrocarbon quantification by GC-MS for the reaction carried out at 5 wt.% of catalyst, 1 h, and biomass:water ratio of 1:2; Figure S9. Hydrolysis reaction of trilaurin; Figure S10. Hydrolysis reaction of monoolein; Figure S11. Hydrolysis reaction of monopalmitin.

**Author Contributions:** Conceptualization, E.F.S.-A.; Data curation, C.Z.C. and D.d.A.A.; Formal analysis, C.Z.C. and J.F.S.d.C.F.; Funding acquisition, E.F.S.-A.; Methodology, E.F.S.-A., C.Z.C., M.A.P.G.C. and D.d.A.A.; Project administration, C.Z.C.; Resources, E.F.S.-A. and M.A.P.G.C.; Supervision, E.F.S.-A. and M.A.P.G.C.; Visualization, D.d.A.A. and J.F.S.d.C.F.; Writing—original draft, E.F.S.-A., C.Z.C. and J.F.S.d.C.F.; Writing—review & editing, E.F.S.-A., C.Z.C., M.A.P.G.C., D.d.A.A. and J.F.S.d.C.F. All authors have read and agreed to the published version of the manuscript.

**Funding:** This research was funded by Fundação Carlos Chagas Filho de Amparo à Pesquisa do Estado do Rio de Janeiro.

**Acknowledgments:** The authors would like to thank Coordenação de Aperfeiçoamento de Pessoal de Nível Superior (CAPES) for the fellowships, and the Fundação Carlos Chagas Filho de Amparo à Pesquisa do Estado do Rio de Janeiro for the funding.

**Conflicts of Interest:** The authors declare no conflict of interest.

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
