# Peer review of "Conversion of Residual Palm Oil into Green Diesel and Biokerosene Fuels under Sub- and Supercritical Conditions Employing Raney Nickel as Catalyst"

_catalysts, doi:10.3390/catal11080995_

Round 1
Reviewer 1 Report
Thermal deoxygenation of palm residue under sub- and supercritical water conditions using Nickel-Raney as heterogeneous catalyst, is presented in this paper. Several experiments were carried out at different reaction temperatures. Furthermore, different times of reaction (1, 3.5, and 6 h) and catalyst loads (5, 7.5, 10 wt.%) were 13 evaluated. However, the following revision is needed before acceptance:
- The abstract should be rewritten. The following aspects should be included in the abstract: A. what is the shortcomings of the other technology reported in the literatures? B. What strategy has been used in the present investigation to overcome the shortcomings of other literatures?
- A relationship between the catalyst’s characteristics and products should be proposed in the mechanism.
- The conclusions are verbose and should be simplified.
Reviewer 2 Report
Please see the file attached.

Reviewer 3 Report
The submitted report is nice and useful basic research on alternative production of liquid parafins (C11 to C17) from residual palm oil following hydrothermal catalytic process using Nickel-Raney catalyst under sub- and supercritical conditions. The main advantage of the discovered and developed procedure is that deoxygenation and decarbonylation processes do not need hydrogen gas loading, since H2 is formed in situ during the process.
I feel that the presented basic study has considerable applied potential and it would be nice if it will be upgraded later with engineering development of the discovered procedure.
The paper is well structured; the experiments properly designed and obtained results adequately evaluated. Overall, the submitted paper is ready to be published as it is.
I can only encourage authors to proceed their efforts with technological evaluation of this basic research.
Round 2
Reviewer 2 Report
Please see the attached file. R. X. refers to the comment on author's response to the particular point (R 1 relates to the Author's response to the Point 1)

Round 3
Reviewer 2 Report
The manuscript is significantly improved and now it is acceptable for publishing if the errors in Figures are corrected (mentioned in first two reviews).
In Figures 11, 13 and 14, authors have "... CO e CO2" conjuntion "e" is not from the English language and it is unacceptable to be in the Figures.
Figure 12 in the third chemical equation still shows 2 mol-equivalents of H2 as the side-product (see first two equations) what is certainly not the case. As a consequence, reader is not sure what a the reactants and what are the products.
In my opinion, these inconsistencies and errors in figures needs to be corrected before publishing.
I emphasised that explicitly in the former review but it was not corrected.
Author Response
We would like to thank the reviewer for giving us the opportunity to once again correct our manuscript. We apologize for not correctly realizing the errors pointed out in previous reviews. As requested, we made corrections to the figures indicated and placed the new figures in the manuscript.